# Determining the Effects of Serial Injections of Pregnant Mare Serum Gonadotropin on Plasma Testosterone Concentrations, Testicular Dynamics, and Semen Production in Leopard Geckos (*Eublepharis macularius*)

**DOI:** 10.3390/ani11092477

**Published:** 2021-08-24

**Authors:** Alexandra K. Mason, Jeongha Lee, Sean M. Perry, Kimberly L. Boykin, Fabio Del Piero, Michael Lierz, Mark A. Mitchell

**Affiliations:** 1Department of Veterinary Clinical Sciences, School of Veterinary Medicine, Louisiana State University, Baton Rouge, LA 70803, USA; mason1@lsu.edu (A.K.M.); kboykin@lsu.edu (K.L.B.); 2Louisiana Animal Disease Diagnostic Laboratory, School of Veterinary Medicine, Louisiana State University, Baton Rouge, LA 70803, USA; jlee282@lsu.edu (J.L.); fdelpiero@lsu.edu (F.D.P.); 3Department of Pathobiological Sciences, School of Veterinary Medicine, Louisiana State University, Baton Rouge, LA 70803, USA; 4Mississippi Aquarium, Gulfport, MS 39501, USA; seanmperry87@gmail.com; 5Clinic for Birds, Reptiles, Amphibians, and Fish, Justus Liebig University Giessen, 35392 Giessen, Germany; michael.lierz@vetmed.uni-giessen.de

**Keywords:** leopard gecko, *Eublepharis macularius*, testosterone, semen, testicle, PMSG, pregnant mare serum gonadotropin

## Abstract

**Simple Summary:**

Current estimates have determined that one in five reptile species are already threatened with extinction. The aim of this study was to determine if pregnant mare serum gonadotropin could be used in leopard geckos (*Eublepharis macularius*) to stimulate the production of sperm and increase their testosterone concentrations. This information may aid in the development of breeding programs for endangered gecko species in the future that will be useful for conservation efforts. Our results demonstrated that this hormone did stimulate sperm production and increased testicular size in leopard geckos. However, it did not increase testosterone concentrations under the current conditions between October–December in the Northern hemisphere. Ultimately, future studies are needed to further characterize the annual reproductive cycle of leopard geckos.

**Abstract:**

Reptiles are highly susceptible to anthropogenic activities as a result of their narrow geographical ranges and habitat specialization, making them a conservation concern. Geckos represent one of the mega-diverse reptile lineages under pressure; however, limited assisted reproductive technologies currently exist for these animals. Exogenous pregnant mare serum gonadotropin (PMSG) has been found to exhibit follicle stimulating hormone-like action and has been routinely used to alter reproductive hormones of vertebrates in assisted reproductive protocols. The purpose of this study was to determine the effects of serial injections of 20 IU and 50 IU PMSG on circulating testosterone concentrations, testicular dynamics, and semen production in a model species of gecko. Twenty-four captive-bred, adult, male leopard geckos (*Eublepharis macularius*) were divided into three treatment groups and administered a once-weekly injection of either PMSG or saline for a total of nine weeks. Ultrasonographic testicular measurements, electrostimulation for semen collection, and venipuncture were performed on days 0, 21, 42, and 63. Right unilateral orchidectomies and epididymectomies were performed in all animals on day 63; tissues were submitted for histopathology. PMSG treated geckos had significantly higher testicular volumes and weights, spermatozoa motility, and spermatozoa concentrations compared with controls. However, there were no significant differences in testosterone concentrations by treatment or time. Under the conditions outlined, PMSG is effective at stimulating spermatogenesis and increasing testicular size, but not effective at increasing testosterone concentrations in the leopard gecko between October–December in the Northern hemisphere.

## 1. Introduction

Geckos are one of three extant mega-diverse lineages of squamate reptiles originating from the major radiations that began diversifying around 200 million years ago [1]. While most gecko species have only been discovered over the last 40 years, as new molecular technologies are developed to aid in our understanding of gecko diversity, many species may already have gone extinct [1]. Modern extinction rates have increased sharply over the past 200 years and correspond to the rise of the industrial society [2]. Reptiles are especially susceptible to anthropogenic threats, including habitat loss and degradation, unsustainable trade, introduced invasive species, environmental pollution, and climate change [3,4,5]. Extinction model estimates that include expected climate changes and habitat losses suggest that as many as 76% of reptiles will be committed to future extinctions by 2050 [6], while current estimates have determined that one in five reptile species are already threatened with extinction [3]. However, due to the rigorous and time intensive process of determining a species’ conservation status, counts of “officially” recognized endangered and threatened species are likely to grossly underestimate the actual number of imperiled species [4].

Reproduction is core to survival, so understanding how an animal breeds is fundamental to conserving species, populations, and, indirectly, the vitality of entire ecosystems [7]. Successful application of assisted reproductive techniques for enhancing propagation, such as artificial insemination, in vitro fertilization, embryo transfer, and germplasm cryobiology, are directly related to the amount of basic reproductive information available for each species [8]. However, despite the bewildering array of reproductive modes exhibited by reptiles, there is little information on the physiology and hormonal control of reproduction for most species [9]. For example, the ability to collect spermatozoa in reptiles is a key initial step in developing assisted reproductive technologies (ART) for these animals; however, in lizards, successful semen collection has only been achieved in nine species through either electrostimulation or manual massage, including: green iguanas (*Iguana iguana*), Grand Cayman blue iguana hybrids (*Cyclura lewisi × nubila*), veiled chameleons (*Chamaeleo calyptratus*), panther chameleons (*Furcifer pardalis*), leopard geckos (*Eublepharis macularis*), Chaco spiny lizards (*Tropodurus spinulosis*), Texas rock lizards (*Sceloporus torquatus*), McCann’s skinks (*Oligosoma maccanni*), and common house geckos (*Hemidactylus frenatus*) [10,11,12,13,14,15,16,17]. Furthermore, the administration of exogenous mammalian gonadotropins to male lizards in order to stimulate steroidogenesis and spermatogenesis has only been attempted in <0.14% (10/6905) of all lizard species [18,19,20,21,22,23,24,25,26,27,28,29,30,31,32]. Prior studies attempting to evaluate the effects of exogenous hormone administration on squamate testes have been terminal, and most of these studies have not examined the effects of the exogenous hormones on circulating testosterone concentrations or semen parameters, which will be paramount in establishing ART in threatened and endangered species in the future.

In mammals, follicle stimulating hormone (FSH) acts at the Sertoli cell to stimulate spermatogenesis, while luteinizing hormone (LH) stimulates steroidogenesis at the level of the Leydig cell. In contrast, a “one gonadotroph, two-cell” theory is generally accepted in reptiles, whereby one gonadotropin, or a gonadotropin complex, is responsible for stimulating both spermatogenesis and steroidogenesis, functions carried out by FSH and LH independently in mammals [19,23,24,33]. In squamates, FSH has been found to stimulate both spermatogenesis and steroidogenesis, while LH has been found to have a similar qualitative effect, although to a lesser degree [21,25,26,28,29,31,34,35,36,37,38]. Exogenous hormones, including pregnant mare serum gonadotropin (PMSG) and human chorionic gonadotropin (hCG), have been previously used with success to develop assisted reproductive programs for mammals, birds, and amphibians. In reptiles, PMSG has been found to have FSH-like activity, whereas hCG has been noted to demonstrate more LH-like activity, although conflicting findings exist on gonadal and endocrine responses to hCG [21,25,26,28,29,31,34,35,36,37,38].

PMSG has been found to be capable of stimulating testicle growth, spermatogenesis, and steroidogenesis in some squamates, including Indian spiny-tailed lizards (*Uromastix hardwicki*) [19,23,39], oriental garden lizards (*Calotes versicolor*) [30], the common agama (*Agama agama*) [21], and the little brown skink (*Leiolopisma laterale*) [24]. In Indian spiny-tailed lizards, PMSG induced marked elevations of both testicular and plasma androgen concentrations [23], and it was several times more potent than hCG at stimulating testosterone synthesis [19]. PMSG was also found to stimulate a four-fold increase in testicular weights and luminal diameter of the testis tubules in common agamas compared with controls, while hCG had a much smaller qualitative effect on testicle size and tubule diameter [21]. Unfortunately, testosterone concentrations were not measured in the common agamas. Male oriental garden lizards administered either hCG, PMSG, or a combination of the two hormones during the quiescent phase were found to have increased testicular weights and diameters, in addition to increased seminiferous tubule diameters and spermatids as the abundant germ cell element [30]. Additionally, cholesterol levels in the testicles were found to be lower in animals receiving PMSG and hCG, suggesting that both exogenous hormones possessed the ability to stimulate spermatogenesis and steroid hormone production in this species of lizard. Exogenous LH, hCG, FSH, FSH + LH, or PMSG administered to the male little brown skinks during the quiescent phase found that all of the mammalian gonadotropins, with the exception of LH, increased the interstitial cell number, stimulated interstitial cell hypertrophy and cytoplasmic granulation, as well as increased epididymal and sexual segment epithelial heights [24]. Based on these findings, PMSG may have FSH-like activity in reptiles, and in some species, FSH stimulation is secondary to preparation of the male reproductive tract with hCG.

The purpose of this study was to determine the effects of serial injections of an exogenous hormone, PMSG, at two different concentrations, 20 IU and 50 IU, on circulating testosterone concentrations, testicular dynamics, and semen production in a model species of gecko, the leopard gecko. The leopard gecko was chosen as a model gecko species due to their abundance, ease of maintaining in captivity, and previously established reproductive seasonality [40]. Refining these techniques in a common species will be important before applying them to threatened or endangered species. Our primary objective was to determine an effective dose of PMSG to increase plasma testosterone concentrations, testicular volumes and weights, the likelihood for semen collection, and spermatozoa motility and concentration with weekly dosing over nine weeks. The hypotheses tested in this study were: (1) PMSG administration would increase testicular volumes and weights; (2) testicular volumes measured on ultrasound would positively correlate with actual testicular volumes; (3) testicular volumes and testosterone production could be effectively determined through non-lethal methods; (4) semen samples could be consistently collected from the leopard geckos by means of electrostimulation; (5) administering exogenous PMSG would increase semen collection success, spermatozoa concentration, and motility; (6) PMSG administration would be associated with a higher prevalence of morphologically normal spermatozoa; (7) PMSG treated geckos would have significantly more histologic changes in the testis and epididymis consistent with spermatogenesis; and (8) PMSG would significantly increase circulating plasma testosterone concentrations.

## 2. Materials and Methods

### 2.1. Ethics Statement

This longitudinal experimental study was performed in accordance with the rules and regulations established by Louisiana State University’s (LSU) institutional animal care and use committee (protocol # 20-043).

### 2.2. Study Species

Twenty-four captive-bred, adult, male leopard geckos were used for this study. The ages and previous sexual histories of the male geckos used in this study are unknown, however, no male in this study had previous contact with females for at least three years while housed at LSU. The sample size was determined using the following a priori information: (1) an alpha = 0.05, a power = 0.80, a 2:1 ratio of PMSG to control animals, an expected difference of >40 pg/mL testosterone concentrations between PMSG and control animals, and a standard deviation (SD) of 20 pg/mL for each group, and (2) an alpha = 0.05, a power = 0.80, a 2:1 ratio of PMSG to control animals, an expected difference of at least 1 × 10^6^ spermatozoa/mL between PMSG and control animals, and a SD of 7.5 × 10^5^ spermatozoa/mL for each group.

### 2.3. Husbandry

Animals were individually housed at Louisiana State University in 43 × 21 × 25 cm clear, plastic containers that were separated with dividers to prevent visualization of other geckos. The environmental temperature range and humidity in the climate-controlled room were 28–29 °C (83–85 °F) and 30–40%, respectively. The geckos were housed on a paper substrate and provided a hiding area and water bowl. A 12-h photoperiod was provided with standard fluorescent lighting. The geckos were fed a diet consisting of gut-loaded and dusted house crickets (*Acheta domesticus*), black soldier fly larvae (*Hermetia illucens*), and mealworms (*Tenebrio molitor*) (Fluker Farms, Port Allen, LA, USA) three times weekly; the amount offered was based on 2% of the geckos’ body weight. Physical examinations were performed on each leopard gecko prior to initiating the study to confirm that they were healthy.

### 2.4. Experimental Design

A prospective experimental study was conducted from October–December (2020); timing was based on the expected non-breeding period for leopard gecko reproduction in the Northern Hemisphere (September–December) [40]. Twenty-four adult male leopard geckos were randomly divided into three treatment groups (group 1: control [saline], *n* = 8; group 2: 20 IU/animal [PMSG], *n* = 8; group 3: 50 IU/animal [PMSG], *n* = 8) using a random number generator (random.org). The chosen dosages of PMSG (pregnant mare serum gonadotropin, sterile filtered white lypolized powder, 1000 IU, ProSpec-Tany TechnoloGene LTD, Rehovot, Israel) were based on previous work performed in reptiles [19,23,39]. A new bottle of PMSG was reconstituted with 2 mL sterile water (Hospira Inc., RL-4428 Lake Forest, IL, USA) to 500 IU/mL for use in the geckos each week, and each animal was administered the appropriate dose subcutaneously over the left epaxial region (shoulder) once weekly for 9 weeks. Control animals received a subcutaneous injection of sterile 0.9% saline at either 0.04 mL or 0.1 mL at the same injection site to mimic the volumes of the 20 IU and 50 IU PMSG doses, respectively. All injections were administered under manual restraint.

All other procedures, including non-invasive testicular measurements, electrostimulation, and venipuncture, were performed under general anesthesia with isoflurane prior to the first PMSG and saline injections (time 0, baseline) and then again once every three weeks (days 21, 42, and 63) of the experiment. Right unilateral orchidectomies with removal of the epididymis were performed in all animals on day 63. The geckos were anesthetized using an induction chamber with 5% isoflurane (Fluriso, VetOne, Boise, ID, USA) and 3 L oxygen/minute. Once the geckos lost their righting reflex, they were removed from the chamber and maintained on 3% isoflurane and 2 L oxygen/minute administered via face mask. Heart rates and respiratory rates were monitored throughout sample collection, with geckos spontaneously ventilating throughout the procedure.

### 2.5. Non-Invasive Testicular Measurement

Testicular measurements of the right testicle were recorded using the Sonoscape S8 (Sonoscape, Centennial, CO, USA) with the 10–15 mHz linear array hockey stick probe as previously described (Figure 1) [12,41]. The right testicle was selected for measurement because it was more easily visualized on ultrasound; visualization of the left testicle was challenging due to superimposition of the gastrointestinal tract. Testicular length and width were recorded on days 0, 21, 42, and 63 of the study (Figure 2). The distance from the cranial to caudal poles of the testicle represented the length. Width was measured in the same view and included the distance from the dorsal to ventral borders of the testicle at its midpoint. Testicular volume was estimated using the following equation: V(mm^3^) = 0.52 LW^2^ [42].

### 2.6. Blood Collection

Whole blood was collected from the ventral tail vein or cranial vena cava of each gecko using a heparinized 25-gauge needle fastened to a 1mL syringe on days 0, 21, 42, and 63 of the study. A total of 0.2 mL whole blood was collected at each time point, ensuring total blood volume collected was <0.8% body weight. Blood samples were placed into lithium heparin microtainers (B-D Vacutainer Systems, Franklin Lakes, NJ, USA) and separated into components using centrifugation at 4000× *g* for 8 min. Plasma was aliquoted into 2 mL cryovials (VWR International, Radnor, PA, USA) and frozen at −80 °C (−112 °F) until it was analyzed for plasma testosterone concentrations.

### 2.7. Electrostimulation/Semen Collection/Semen Evaluation

Semen was collected using electrostimulation on days 0, 21, 42, and 63 of the study. The vent and cloaca were cleaned with a Kimwipe (Kimberly-Clark Professional, Corinth, MO, USA) to remove debris. While anesthetized, each animal was electrostimulated using a 360° circumferential metallic probe (20 mm length, 3 mm diameter) connected to a variable amperage power source [13,41,43]. An intromission was defined as the process of fully inserting the metallic portion of the probe into the vent and directing it cranially. Based on ultrasound measurements, the probe length was sufficient to reach the caudal pole of the testicles. Animals were electrostimulated by performing three series of intromissions: 15 intromissions at 0.1 mAmps, 15 cloacal intermissions at 0.15 mAmps, and 15 intromissions at 0.2 mAmps. A three-minute break was provided in between each series of intromissions. Electrostimulation was discontinued following the collection of a semen sample. Any fluid observed in the cloaca following a series of intromissions was collected with a 2–20 µL single channel pipettor. Each sample was evaluated for the presence or absence of spermatozoa by placing the fluid directly on a glass slide with a cover slip and reviewing it under light microscopy (100× and 400×) at ambient temperature. If spermatozoa were visualized, the intromission number required for successful collection of an ejaculate, in addition to semen color and volume collected, were recorded. Motility of spermatozoa was determined by estimating the percentage of progressively motile spermatozoa to the nearest 5% in 5 high powered fields (magnification, 400×). All motilities were measured by a single reviewer (MM). The sample on the microscope slide and coverslip were washed into a 2 mL microcentrofuge tube (VWR International, Radnor, PA, USA) with a 1:40 dilution of formal saline. Spermatozoa concentration was determined with a Neubauer hemocytometer with phase contrast microscopy (magnification, 400×). Spermatozoa were counted in all 25 cells on each side of the hemocytometer and the total number of spermatozoa calculated by multiplying by the dilution factor. Spermatozoa morphology were recorded for each sample when at least 50 spermatozoa could be assessed. The number of morphologically normal spermatozoa, in addition to sperm with folded tails, kinked midbodies, detached heads, retained proximal droplets, coiled tails, head defects, and distal droplets were evaluated. All spermatozoa counts and morphology were performed by a single reviewer (AM).

### 2.8. Unilateral Orchidectomy and Epididymectomy Procedures

On day 63 of the study, all twenty-four geckos underwent a surgical procedure to remove their right testicle and epididymis for morphometric measurements and gross and histopathological assessment. Each gecko was already anesthetized with isoflurane inhalant gas for venipuncture, ultrasound, and electrostimulation, and was administered subcutaneous injections of dexmedetomidine (Zoetis Services LLC, Parsippany, NJ, USA) 0.025 mg/kg, hydromorphone (Hospira, Inc., Lake Forest, IL, USA) 0.5 mg/kg, and meloxicam (OstiLox, VetOne, Boise, ID, USA) 0.3 mg/kg for additional sedation and analgesia just prior to surgery. Anesthesia was monitored throughout the procedure by measuring the respiratory rate, Doppler heart rate, and presence/absence of muscle tone and reflexes. The geckos were placed in dorsal recumbency, and their surgical site (ventral right abdominal region) was aseptically prepared with chlorohexidine scrub (VetOne, Boise, ID, USA) and 0.9% sterile saline. A #11 scalpel blade (Bard-Parker, Aspen Surgical Products, Inc., Caledonia, MI, USA) was used to make an initial paramedian incision on the right side of the abdomen, and Metzenbaum scissors were used to extend the body wall incision (3–4 cm). A Lone Star self-retaining retractor (Cooper Surgical Inc., Trumbull, CT, USA) was used to enhance visualization within the coelomic cavity. The ventral aspect of the intra-abdominal fat pad was immediately visualized upon entering the coelomic cavity. Gentle retraction of the fat pad revealed that the thin-walled urinary bladder was adhered to the dorsal wall of the fat pad. Medial displacement of the intestines using a cotton tipped applicator (Puritan Medical Products, Guilford, ME, USA) revealed the right testicle (Figure 3) and epididymis along the dorsal body wall (Figure 4). Once in the visual field, the thin mesorchium at the cranial pole of the testicle was gently grasped with atraumatic forceps to aid in the exteriorization and visualization of the testicle. A small hemoclip (Titanium ligating clips, Weck, Morrisville, NC, USA) was placed on the testicular artery and veins to control hemostasis, and the testicle was dissected from the remainder of the mesorchium for removal. The remaining epididymis was then grasped at its cranial end and traced caudally so an additional hemoclip could be placed and the epididymis removed. Unfortunately, it was challenging to remove the testicle and epididymis en bloc. Sterile cotton tip applicators were used to apply pressure for additional hemostasis as necessary, and the abdomen was flushed with sterile saline prior to closure. The body wall was closed with 4-0 Maxon (Coviden, Mansfield, MA, USA) in a continuous pattern, and the skin was also closed with 4-0 Maxon using a horizontal mattress pattern. Sterile skin glue (GLUture, Zoetis, Kalamazoo, MI, USA) was applied to the incision to reduce seepage. A subcutaneous injection of atipamezole (Zoetis Services LLC, Parsippany, NJ, USA) 0.5 mg/kg was administered to reverse the dexmedetomidine. The geckos were monitored post-operatively until all reflexes had returned and they were able to ambulate normally. Each animal received an additional injection of hydromorphone 0.5 mg/kg subcutaneously the following day, in addition to 0.3 mg/kg meloxicam subcutaneously once daily for three consecutive days to minimize discomfort. Animals were observed daily for 6 weeks, post-operatively, for any negative side effects associated with the surgical procedure, including anorexia, depression, discharge or swelling at the incision site, dehiscence, and lack of energy or ambulation.

### 2.9. Gross and Microscopic Assessment of the Reproductive Tract

Testicular and epididymal tissues were immediately rinsed with sterile 0.9% saline upon removal from the body cavity and pat dried using a Kimwipe. The testicle and epididymis were weighed separately to the nearest milligram using an analytical balance, and testicle length and width were measured using digital calipers. A gonadosomatic index (GSI) was calculated using the following formula: (testicle weight [g]/body weight [g]) × 100. Snout-vent length (SVL) and snout-tail length (STL) were also obtained while the geckos were anesthetized.

The testicles and epididymides were fixed in 10% neutral buffered formalin, routinely processed, and embedded in paraffin, and 5 μm sections were stained with hematoxylin and eosin for histological analysis. Germ cell identification was performed under light microscopy by reviewing five sections of seminiferous tubules per animal. For micrometric measurements, the slides were scanned using a digital slide scanner (Nanozoomer C9600-02, Hamamatsu Photonics, Hamamatsu City, Japan). The measurements were taken using Aperio ImageScope software (Leica Biosystems, Buffalo Grove, IL, USA). Five measurements were used to define the diameter of the epididymis; ten measurements to define epididymal epithelial height; twenty measurements to define the diameter of the seminiferous tubules; five measurements to define the numbers of interstitial cells; ten measurements to define the interstitial cell nuclear diameter from each animal. The numbers of interstitial cells were counted from triangular interstitial areas formed by three sections of seminiferous tubules. For the diameter of seminiferous tubules, diameters of round sections, or short axes of elongated sections were measured. Intraepithelial secretory granules of the epididymal epithelial cells, intraluminal spermatozoa in the epididymis, interstitial cell cytoplasm, and vacuolation in the testes were graded from 1 to 3. All samples were reviewed by the same author (JL).

### 2.10. Testosterone Assay

An enzyme immunoassay kit (EIA) (Arbor Assay DetectX Testosterone K032-H5, Ann Arbor, MI, USA) was used to measure plasma testosterone concentrations. This assay has been previously validated by the authors in leopard geckos, and the same methods were followed in the present study [32]. All samples were processed at the conclusion of the study. The published sensitivity for this assay is 9.92 pg/mL, with a limit of detection at 30.6 pg/mL. Based on previous validation, samples were diluted at 1:20 [32]. To determine the repeatability of the assay, intra- and inter-assay coefficients of variation (CV) were measured. Intra-assay CV was measured by examining the CV of each sample run in duplicate, while inter-assay CV was measured by analyzing the same samples on different plates. Values with CV < 15% were considered data, while samples with CV > 15% were re-analyzed.

### 2.11. Statistical Analysis

The distributions of the data were evaluated using the Shapiro–Wilk test, skewness, kurtosis, and q-q plots. Data that did not meet the assumption of normality were log transformed for parametric testing. Data that were normally distributed are reported by the mean, SD, and minimum-maximum values (min–max), while non-normally distributed data are reported by the median, 25–75%, and min–max. Mixed linear models were used to determine if there were differences in the body weight, spermatozoa concentrations, testicular volume, ejaculate volume, testosterone concentrations, and spermatozoa motility by time and treatment. Leopard gecko was included in the model as the random variable, while time and treatment were fixed factors. Separate models were created for spermatozoa concentration to evaluate the treatment variable with three levels (saline, 20 IU, 50 IU) and two levels (saline, PMSG). One-way analysis of variance (ANOVA) testing was used to determine if post-surgical testicular volume, weight, or GSI differed between the treatment groups and the controls. Least significant difference tests were used for any post-hoc comparisons if the ANOVA was significant. Levene’s test was used to assess for homogeneity of variance. If no difference was noted when the three levels of treatment were compared, an independent samples t-test was used to determine if differences existed between the controls and PMSG treated animals (20 IU and 50 IU animals combined). One-tail testing was used for these comparisons. The same analyses were used to determine if seminiferous tubule diameter, epithelial height, interstitial cell nuclear diameter, interstitial cell number, and epididymal diameter differed by treatment groups. Kruskal–Wallis tests were used to determine if secretory granule content, intraluminal spermatozoa, interstitial cell cytoplasm, and cytoplasmic vacuoles on post-surgical testicles differed by treatment groups. If not significant when comparing all three groups, a Mann–Whitney test was used to make the same comparisons for saline versus PMSG combined treated geckos. A generalized linear model for an ordinal logistic response was used to determine if the number of intromissions required to collect a semen sample differed by treatment group or time; the same model was used to determine if the number of mature spermatozoa differed by treatment group and sections samples. Generalized linear models for linear responses were used to determine if spermatozoa morphologic characteristics differed by treatment or time. Pearson’s correlation coefficients were calculated to determine if body weight, testicular weight and volume, and SVL were correlated. SPSS 24.0 (IBM Statistics, Armonk, NY, USA) was used to analyze the data. A *p* ≤ 0.05 was used to determine significance.

## 3. Results

There was a significant difference in body weight over time (F = 7.94, *p* < 0.0001), but not by treatment (F = 1.32, *p* = 0.284) or interaction of treatment and time (F = 0.33, *p* = 0.888). Body weights on day 63 were significantly lower than baseline (*p* < 0.0001), 21 days (*p* < 0.0001), and 42 days (*p* = 0.012); there was no difference in body weights between the other sampling periods (all *p* > 0.081) (Table 1). Body weight was positively correlated with snout vent length (R: 0.534, *p* = 0.007), but not snout tail length (R: −0.09, *p* = 0.673).

There was a significant difference in testicular volume measured by ultrasound by treatment (F = 4.62, *p* = 0.006) and time (F = 3.98, *p* = 0.034) but not the interaction of treatment and time (F = 0.73, *p* = 0.625). Testicular volumes by ultrasound were significantly larger (*p* = 0.011) in the 20 IU (median: 63.62, 25–75%: 49.14–76.14, min–max: 20.10–141.52) treatment group compared to baseline (median: 38.47, 25–75%:25.50–53.68, min–max: 13.05–81.47). There was no significant difference in testicular volume by ultrasound between the 50 IU (median: 52.77, 25–75%: 35.15–70.07, min–max: 14.72–122.91) and control groups (*p* = 0.107) or the 50 IU and 20 IU groups (*p* = 0.272). Testicular volumes by ultrasound were significantly larger on day 63 compared to baseline (*p* < 0.0001) and day 21 (*p* = 0.04), and day 42 testicular volumes were significantly higher than baseline (*p* = 0.04) (Table 2).

There was a significant positive correlation between post-surgical testicular volume and testicular weight (R: 0.936, *p* < 0.001); however, there was no correlation between post-surgical testicular weight and body weight (R: 0.014, *p* = 0.949) or SVL (R: 0.095, *p* = 0.659). There were significant differences in post-surgical testicular volume (F = 3.53, *p* = 0.024) and testicular weight (F = 2.76, *p* = 0.043) by treatment group. The saline group had significantly smaller post-surgical testicular volumes (20 IU, *p* = 0.009, 50 IU, *p* = 0.036) and weights (20 IU, *p* = 0.018, 50 IU, *p* = 0.043) compared with the PMSG treated animals (Table 3). There were no significant differences in post-surgical testicular volumes (*p* = 0.243) or weights (*p* = 0.314) between the PMSG treated geckos. The GSI was significantly higher (F = 4.1, *p* = 0.028) in the PMSG (mean ± SD: 0.13 ± 0.05, min–max: 0.06–0.24) treated geckos compared with the control geckos (mean ± SD: 0.08 ± 0.06, min–max: 0.01–0.18).

Testicular volume by ultrasound was positively correlated (R: 0.672, *p* < 0.001) to post-surgical testicular volume. Post-surgical testicular volume was always higher than ultrasound measured testicular volume, except for one case. There was no significant difference in testicular volume difference between treatments (F = 0.923, *p* = 0.413). On average, post-surgical testicular volumes were 1.7 ± 0.62 (min–max: 0.13–2.54) times larger than ultrasound measured testicular volumes.

Electrostimulation was successful in 85.4% (82/96) of the cases over 63 days. Semen was not collected in 6/32 (18.7%) events for the control and for 20 IU groups, while only 2/32 (6.2%) attempts in the 50IU group were unsuccessful. The majority (10/14, 71.4%) of the unsuccessful electrostimulation events were during the baseline sampling; 3/14 (21.4%) and 1/14 (7.1%) negative events were from the 21 and 42 day sampling periods, respectively. There was no significant difference in the number of intromissions required to collect semen by treatment (Χ^2^ = 1.3, *p* = 0.529) or time (Χ^2^ = 2.4, *p* = 0.502). There was no difference in ejaculate volume by treatment (F = 0.34, *p* = 0.718), time (F = 1.25, *p* = 0.296) or the interaction of time and treatment (F = 0.24, *p* = 0.961). Because there were no differences in ejaculate volume, the data were combined for a single reference (median: 2.0 µL, 25–75%: 2.0–2.0, min–max: 0–5.0).

There was a significant difference in motility by treatment (F = 4.89, *p* = 0.018) and time (F = 4.7, *p* = 0.014). The interaction of treatment by time was not significant (F = 0.958, *p* = 0.477). Motility was significantly higher in the 20 IU (*p* = 0.006) and 50 IU (*p* = 0.049) groups compared with the controls (Table 4). There was no significant difference in motility between the two PMSG groups (*p* = 0.345). Spermatozoa motility was found to be significantly higher in the 42 and 63 day sampling periods compared to baseline and 21 day samples (Table 5).

There was a significant difference in spermatozoa concentrations over time (F = 6.7, *p* = 0.002) and for the interaction of time by treatment (F = 2.5, *p* = 0.054). There was no significant difference in spermatozoa counts by treatment (F = 3.25, *p* = 0.064) when evaluating all three treatments (saline, 20 IU, 50 IU), but it approached significance. Spermatozoa counts were significantly higher at 42 (baseline, *p* = 0.009; 21 days, *p* = 0.002) and 63 days (baseline, *p* = 0.006; 21 days, *p* = 0.004) compared to baseline and 21 days (Table 6). There were no significant differences in spermatozoa counts between baseline and 21 days (*p* = 0.703) or 42 and 63 days (*p* = 0.749) (Table 6). When evaluating the model with the treatment variable at two levels (saline, PMSG [combined 20 IU, 50 IU]), treatment (F = 4.34, *p* = 0.042) and treatment by time (F = 4.35, *p* = 0.038) were found to be significantly different, while time was not (F = 2.65. *p* = 0.114). Spermatozoa concentrations were significantly higher in the PMSG treated animals (*p* = 0.042) compared to the saline treated animals (Table 6).

The presence of normal spermatozoa was not impacted by time (X^2^ = 0.8, *p* = 0.845), but was significantly different by treatment (X^2^ = 10.0, *p* = 0.007) (Table 7). There were significant differences in the presence of folded tails (X^2^ = 22.8, *p* < 0.001) and kinked midbodies (X^2^ = 22.9, *p* < 0.001) by time, but not by treatment (folded tail: X^2^ = 0.7, *p* = 0.698; kinked midbody: X^2^ = 0.8, *p* = 0.661) (Table 8). There was no significant difference in the likelihood of distal droplets (time: X^2^ = 1.4, *p* = 0.569; treatment: X^2^ = 0.4, *p* = 0.829), head defects (time: X^2^ = 1.5, *p* = 0.523; treatment: X^2^ = 2.3, *p* = 0.313), detached heads (time: X^2^ = 2.8, *p* = 0.422; treatment: X^2^ = 0.9, *p* = 0.623), retained proximal droplets (time: X^2^ = 5.2, *p* = 0.157; treatment: X^2^ = 2.1, *p* = 0.345), or coiled tails (time: X^2^ = 2.9, *p* = 0.393; treatment: X^2^ = 1.1, *p* = 0.561) by time or treatment group (Table 9).

There was a significant difference in the seminiferous tubule diameters (F = 4.4, *p* = 0.025) by treatment group, with PMSG treated gecko seminiferous tubule diameters being higher (mean: 229.9 µm, SD: 16.8, min–max: 200.5–264.9) than saline controls (mean: 205.6, SD: 40.9, 135.1–255.6). There were no significant differences in epididymal diameter (F = 0.9, *p* = 0.356), interstitial cell nuclear diameter (F = 0.07, *p* = 0.785), interstitial cell number (F = 0.629, *p* = 0.436), or epididymal epithelial height (F = 2.5, *p* = 0.067) between saline and PMSG treated geckos, although epididymal epithelial height approached significance. There was a significant difference in epididymal intraluminal spermatozoa (z = −1.6, *p* = 0.045) between treatment groups, with abundant spermatozoa found in 78.6% (11/14) of PMSG treated geckos and only 40% (2/5) of saline treated geckos. There were no significant differences in secretory granule content (Z = −0.394, *p* = 0.347) between treatment groups. There were no significant differences in cytoplasmic vacuolization (Z = −2.43, *p* = 0.417) or interstitial cell cytoplasm (Z = −2.57, *p* = 0.417) between saline and PMSG treated geckos. There was no significant difference in the presence of mature spermatozoa in the post-surgical testicles based on sample section reviewed (X^2^ = 3.7, *p* = 0.491) or treatment group (X^2^ = 3.9, *p* = 0.142). Mature spermatozoa were found to be abundant (80.8%) in the post-surgical testicles; fewer samples were found to have moderate (10.8%), few (6.7%), or absent (1.7%) mature spermatozoa. Spermatogonia, round spermatids, elongate spermatids, and primary spermatocytes were present in all three groups of geckos.

There was no significant difference in testosterone concentrations over time (F = 2.1, *p* = 0.139), treatment group (F = 0.703, *p* = 0.507), or the interaction of time and treatment group (F = 1.6, *p* = 0.220). Because there was no difference in time or treatment groups, leopard gecko testosterone reference intervals (Table 10) were established for the months of October, November, and December according to the American Society of Veterinary Clinical Pathologists [44]. The Tukey’s test was used to screen for outliers [44]. For October, there were two outliers: gecko 19 (358.2 ng/mL) and gecko 4 (524.2 ng/mL); these data were removed for reference interval determination. There were no outliers for November or December. MedCalc 17 (MedCalc Software, Ostend, Belgium) was used to determine the central 95th percentile of the data. Because the data were not normally distributed, the central 95th percentiles of non-normally distributed data were determined using non-parametric methods established by the Clinical and Laboratory Standards Institute [45]. Confidence intervals of 90%, for the lower and upper limits, could not be determined for this data set due to the number of samples being less than 120.

## 4. Discussion

The results of this study confirmed the majority of the authors’ original hypotheses, except two: that animals administered PMSG would have histological changes associated with increased testosterone production and that there would be higher circulating plasma testosterone concentrations in PMSG treated animals compared to controls. Administration of PMSG was found to increase testicular volume and weight, and final testicular volumes measured on ultrasound were positively correlated with actual testicular volumes. Testicular volume and circulating testosterone concentrations were determined non-lethally in this model species of gecko, giving hope for future conservation programs with threatened and endangered species. Electrostimulation was determined to be an effective method to collect semen repeatedly in leopard geckos, and semen collection, spermatozoa concentrations, and motility all increased over time in PMSG treated geckos. While seminiferous tubule diameters were significantly increased in PMSG animals, indicating that there was a gonadal effect on spermatogenesis, there were no differences in interstitial cell number, nuclear diameter, cytoplasmic vacuolation, amount of cytoplasm, or secretory granule content in the epididymal epithelium. The results of this study confirm that PMSG can have a direct impact on the male leopard gecko reproductive tract from October–December in the Northern Hemisphere. The ability to obtain pharmacological control over the reproductive system of geckos will enable scientists to manipulate the reproductive cycle to reduce dependency on natural breeding seasons in these animals.

Gecko body weights decreased over time. The diet offered to the geckos remained constant over the course of the study; thus, the weight loss noted could not be attributed to access to energy but instead some other factor(s). Physiologic stress, characterized by an increase in glucocorticoid synthesis and catabolism of stored energy, can be associated with routine restraint and handling [46,47,48,49]. The authors’ attempted to reduce gecko handling over the course of the study by limiting injections to once weekly and sampling frequency to once every three weeks. It is possible that a shorter study with more frequent dosing could reduce overall handling and should be considered in the future. Additionally, while behavior was not monitored over the course of this study, wild male reptiles often migrate short distances for breeding purposes [50], which can be energetically costly. In the authors’ experience (SMP, MAM), it is not uncommon for captive male squamates to reduce food consumption during the breeding season. Based on the high testosterone concentrations measured in all geckos, regardless of treatment, it is possible that the weight loss was attributed to the prenuptial reproductive cycle of the geckos. Increasing the availability of food during periods of increased reproductive activity may help offset weight loss; however, if these animals have reduced food consumption for physiologic reasons, we may just need to expect weight loss during this period of the reproductive cycle. Results of this study also indicate that gecko body weight and SVL are correlated, but that body weight and STL were not correlated. This discrepancy is likely due to the differences in tail length in these animals, and possible variation between normal and re-grown tails. Thus, SVL is a better indication of weight than STL in leopard geckos.

PMSG increased testicular sizes in treatment animals compared to those administered saline, as measured by elevated GSI, testicular weights, ultrasound and post-operative testicular volumes, and correlating with a higher degree of sperm production in PMSG animals. Spermatozoa are produced in the testicles, and testis size in reptiles is maximal at the time of spermiogenesis, suggesting that large testes are indicative of a high spermatozoa production at the individual level [51]. In the common agama, a four-fold increase in GSI (mean GSI 0.88) was observed in animals receiving PMSG compared to control animals after 21 days [21]. While the difference in GSI was not as dramatic in the leopard geckos, a nearly two-fold increase was observed in animals administered PMSG compared to controls, suggesting that PMSG administration had a significant impact on spermatogenesis.

PMSG administered to leopard geckos was also successful at increasing testicular volumes as measured by ultrasound over time, but these results were not dependent on the dosage of PMSG administered. Other studies, evaluating the use of PMSG in lizards, did not vary their doses of PMSG in order to determine their effects [19,21,23,24,30,39], however, PMSG was found to increase testicle size and promote spermatogenesis in these studies. Effective dosages of PMSG in lizards have ranged from 1 IU in *Leiolopisma laterale* [24] to 100 IU in the *Agama agama* [21] and demonstrated histologic changes at the level of the testis and epididymis. Thus, the effects of PMSG in lizards may not be dose dependent. While no current attempt has been made to standardize dosing, standardization will be necessary in order to develop functional reproductive programs in the future.

Testicular volume as measured on ultrasound was positively correlated to the post-surgical testicular volume, suggesting that ultrasound is a viable option for non-invasive monitoring of testicular volume in a small lizard species. Ultrasound was initially selected by the authors as a non-invasive method for measuring testicle size to develop a clinical, ante-mortem method to assess the reproductive cycles of male reptiles. Previous studies evaluating the effects of exogenous hormone administration on testicle size and function have relied on post-mortem measurements to determine their effectiveness [19,21,22,23,24,28,29,30,31,38,39,52,53]. By using ultrasound, it is possible to conduct these types of studies on threatened and endangered species. The equation used in this study to estimate testicular volume was based on the volume of an ellipsoid [42] and only requires two measurements (length and width) to estimate volume. This was advantageous because obtaining a second image to evaluate testicle width was challenging in these small lizards. This method for obtaining non-invasive testicular measurements was also found to be effective in veiled and panther chameleons [41]. However, in contrast to the chameleon study, the right testicles of all leopard geckos were removed, weighed, and measured to calculate actual testicular volumes, allowing for the validation of this equation and comparison between gross and ultrasound measured testicles. Actual testicular volumes were found to be higher than ultrasound measured testicular volumes (except in one animal); thus, ultrasound measurements may underestimate actual volume. Reptile testes are intracoelomic, elongated, and cylindrical in shape [54]; lie dorsocaudal to the liver; are suspended by the mesorchium [55]. Due to their position in the body cavity, accurate identification of the testicle borders may be reduced by the super-imposition of the gastrointestinal tract, urinary bladder, liver, and intra-abdominal fat pads. Initially, the authors of this study had planned to measure both testicles via ultrasound in each animal, and then randomly remove either the left or right testicle for histopathologic assessment. However, the left testicle was often difficult to visualize due to interference from the gastrointestinal tract, thus, the more consistently visualized right testicle was selected for routine measurement and histologic assessment in all geckos.

While a previous study successfully used electrostimulation in the leopard gecko to collect semen samples at a single time point [13], the results of the current study confirm that electrostimulation is a safe and effective means of reliably collecting repeated semen samples from leopard geckos over time. Electrostimulation was successful in producing semen samples in 85.4% of the attempts made in both the control and treatment gecko groups. Electrostimulation of Texas rock lizards (*Sceloporus torquatus*), Chaco spiny lizards (*Tropidurus spinosus*), and green iguanas produced similar successes, with semen collected in 77%, 94%, and 88% of samples collected, respectively [10,14,15]. Lower results were reported for panther (55%) and veiled (50%) chameleons [12]. Sampling was performed during the non-breeding season for the geckos, but during the breeding seasons of the other lizards. These findings affirm that electrostimulation can be used to collect semen from leopard geckos with a high degree of success during the non-breeding season regardless of additional exogenous treatments.

Unlike mammals, the neuronal pathway that controls ejaculation in reptiles is unknown, but anatomic similarities between the urogenital systems of mammals and reptiles suggest they have similar innervations [10]. Additionally, a recent study in a porcine model suggested that electrostimulation directly activates pelvic musculature rather than neural mechanisms [56]. While the results obtained in the leopard geckos and other lizards support this idea, more studies are ultimately needed to confirm the underlying mechanisms.

Isoflurane anesthesia was used in this study due to the perceived discomfort associated with electrostimulation [10,41]. While cattle routinely undergo electrostimulation without anesthesia, and electrostimulation was successfully utilized in the spiny lava lizard without sedation [14], the authors find that the significant amount of muscle contraction associated with the procedure must cause some discomfort. Additionally, in birds, it was determined that while there are perceived intra- and interspecific behavioral variations to electric impulses, the second intromission series of electrostimulation was found to induce much more agitation and vocalization in sampled males than did the first series of intromissions [57]. Follow-up studies attempting to objectively measure the degree of discomfort in reptiles are warranted. When considering anesthetics for this type of procedure, it is important to consider potential sequelae to treatment. For example, opioids have been associated with adverse effects on spermatozoa and should be avoided [58,59]. The authors only used an opioid, hydromorphone, after the final semen collection (day 63) and immediately prior to the orchidectomy for pre-emptive analgesia to limit any impact on spermatozoa. More studies are needed to determine the potential impacts of anesthetics and analgesics on reptile spermatozoa.

Spermatozoa concentration and motility were found to increase over time and by treatment during the course of this study. However, no difference in ejaculate volume was observed based on treatment or time. PMSG has been found to possess FSH-like activity, promoting spermatogenesis in some lizard species [21,23,24,30,39]. Additionally, reptile spermatogenesis may take 5–8 weeks to complete [60]. The results of the present study coincide with these previous findings. Spermatozoa concentrations and motility increased over time in the PMSG treated geckos, with both being significantly higher six (42 days) and nine weeks (63 days) after initiating treatment. The initial median motility of spermatozoa collected at baseline was 0%, but this improved to 45% at day 63 (9 weeks) of the experiment. In other lizard species, spermatozoa motilities were found to be 78% in both green iguanas and spiny lava lizards, 70% in McCann’s skinks, 51–93% in the tegus (*Tupinambis merianae*), and 0–100% in veiled and panther chameleons [10,12,14,16]. While the potential mechanisms behind PMSG activity on spermatozoa motility are not currently understood, motility is a trait of mature sperm and may be impacted by differences in anatomy and physiology. In lizards, spermatozoa pass from the epididymis into the ductus deferens and gain maximum motility in the distal segment where they accumulate; testicular spermatozoa have poor motility (1%) [61]. PMSG may have a secondary impact on increasing leopard gecko spermatozoa motility as they undergo the maturation process and are expelled. The effects of PMSG on leopard gecko spermatozoa concentration and motility were not dose dependent, and future studies may aim to use the lowest effective dose of exogenous gonadotrophin to elicit an effect. Some authors believe that reptile spermatozoa motility is tied to some induction agent [62], and thus may be why higher average motilities have been observed in other studies that have taken place during normal breeding seasons.

Median ejaculate volume in this study was 2.0 µL. Ejaculate volumes of the leopard gecko were similar to those obtained in veiled (2.0 µL) and panther chameleons (2.9 µL), but were lower than those observed in other studies (4.6 µL in the Texas rock lizard) [12,15]. However, even larger lizards, such as the green iguana, produce small ejaculate volumes (median 50 µL) [10]. Low semen volumes in reptiles have been associated with a lack of accessory sex glands [10]. However, despite the low ejaculate volumes, the concentration and motility of spermatozoa did not appear to be adversely affected. The small volumes can limit the number of tests that can be done to evaluate the sample; however, extending the sample to increase sample volume can help overcome this deficiency [10].

There were more morphologically normal spermatozoa in PMSG treated leopard geckos compared to the control animals. Folded tails were the most common morphological defect, and became more prevalent over time, while kinked midbodies were the second most common anomaly, but improved over time. The electrostimulation technique used in this study was similar to that employed in green iguanas, leopard geckos, and chameleons [10,13,41]. However, how electrostimulation affects the male lizard reproductive tract, and where the semen is dispelled from, is unknown. Manual manipulation techniques, such as those employed in the New Zealand gecko [17], or a combination of manual manipulation and electrostimulation may be more successful at obtaining a physiologic ejaculate with less morphologic defects and contamination in the leopard gecko. Other squamate species have been found to have higher proportions of morphologically normal spermatozoa than were observed in the leopard geckos. In the Chaco spiny lizard, no morphological abnormalities were observed in semen samples [14], whereas green iguanas, corn snakes (*Pantherophis guttatus*), veiled chameleons, and panther chameleons had 94%, 75%, 56.5%, and 55% morphologically normal sperm, respectively [10,12,63]. However, the majority of these studies were performed during the breeding season of the species examined, when the normal spermatogenic cycle occurs, and morphologically normal spermatozoa would be expected. Despite expected elevations in testosterone concentrations amongst leopard geckos, the authors suspect they had not officially entered into the breeding season yet (January–September, Northern Hemisphere), and their reproductive tracts were still being primed prior to spermatogenesis. Thus, electrostimulation may have expelled spermatozoa from the epididymides prior to complete maturation, resulting in a larger proportion of secondary morphological defects. The higher proportion of morphologically normal spermatozoa in PMSG treated animals further supports the authors’ theory that this hormone has FSH-like activity in leopard geckos and is capable of stimulating spermatogenesis in these animals outside of their normal breeding period.

The histological findings of this study confirm that PMSG had an impact on spermatogenesis, but did not significantly impact steroidogenesis. The measurements collected in this study, including the diameters of the epididymis and seminiferous tubules, presence of germ cell developmental stages, epididymal epithelial heights, numbers of epididymal intraepithelial secretory granules, numbers of interstitial cells, and abundance of interstitial cell cytoplasm and cytoplasmic vacuoles, were based on previous histologic descriptions in other lizards [24,30,60]. Seminiferous tubule diameters were larger in animals receiving PMSG compared with controls. Additionally, intraluminal spermatozoa in the epididymides were nearly twice as abundant in animals receiving PMSG (Figure 5) compared to saline (Figure 6). However, there were no significant differences in the interstitial cell numbers or nuclear diameters, nor the epidydimal epithelial heights between the saline and PMSG treatment groups, suggesting that no additional Leydig cell hyperplasia or hypertrophy took place in animals receiving PMSG. The findings of this study are in contrast to others in reptiles that directly measured testosterone concentrations and testicular histology following the administration of PMSG [19,21,23,24,30]. In these previous studies, significant increases in circulating testosterone were measured following PMSG administration, and histology of the testes and epididymides noted an increase in the interstitial cells, suggesting probable androgen production. However, in these studies, sampling was conducted during the quiescent phases of reproduction for each species. The breeding season of leopard geckos in the Northern Hemisphere begins as early as January and extends to late September [40]. However, despite this study taking place firmly within the proposed non-breeding phase for this species, the animals used in this study possessed high mean baseline testosterone concentrations averaged over the three months of the study (79.4 ng/mL). Based on these results, it is possible that the leopard gecko follows a three-phase reproductive cycle, similar to those exhibited in the Caspian bent-toed gecko (*Cyrtopodion caspium*) and the house gecko (*Hemidactylus flaviviridis*). However, it is also possible that, similar to the common gecko (*Hemidactylus brooki*), leopard geckos may be spermatogenically active throughout the year [64]. The seasonally breeding Caspian bent-toed gecko was described as having three phases of spermatogenesis: the active, transitional, and inactive phases [65]. Additionally, house geckos possess a three-phase reproductive cycle, characterized by quiescent, recrudescent, and active phases [66]. The quiescent phase of the reproductive cycle in the house gecko was characterized by flaccid, small testes, no spermatogenic activity, and low plasma steroid concentrations, while the recrudescent phase, occurring in September–October, demonstrated increasing testicular mass, increased primary and secondary spermatids, a rise in plasma steroid concentrations, and increased steroidogenic factors in the Sertoli and Leydig cells. The active phase, occurring in November–May, exhibited large numbers of mature spermatozoa in all sections of the epididymis, peak plasma steroids, and fully developed ultrastructural steroidogenic features [66]. Conversely, previous findings in common geckos from India determined that, although there was significant variation in testicular mass between different months of the year, the testes were spermatogenically active throughout most of the year, with the exception between June and September, during the wet season, where few animals possessed abundant spermatozoa [64]. Smaller sizes of Leydig cell nuclei were also observed from May–August in these geckos, suggesting reduced androgen output during low spermatogenic activity. Lizards, in general, exhibit prenuptial spermatogenesis [64], with spermatozoa being produced prior to mating. Thus, PMSG likely did not further increase testosterone production during this period of early spermatogenesis since it was already elevated naturally during this phase, further supporting the idea that leopard geckos likely follow a prenuptial pattern of reproduction. Based on results obtained in the current study, the authors propose that the leopard geckos were either in the late stages of the recrudescent phase of spermatogenesis, or they may exhibit more continuous spermatogenesis throughout the year, and PMSG acted to further stimulate spermatogenesis in these animals. Ultimately, extending sample collection from January through September will be needed to confirm one of these theories.

The mean testosterone concentrations measured in the leopard geckos during this study were higher than baseline testosterone concentrations recorded in other lizards, including veiled (12.93 ng/mL) and panther chameleons (11.64 ng/mL) [12], house geckos (15 ng/mL) [66], and green iguanas during their reproductive season (29.7 ng mL^−1^) [67]. Additionally, the range of testosterone concentrations observed in the leopard geckos (11.1–465.7 ng/mL) varied widely and were not dissimilar to previously reported mean testosterone concentrations for this species (87.6–139.69 ng/mL) and Madagascar ground geckos (*Paroedura picta*) (3.30 to 144.22 ng/mL) [68,69]. Thus, some species of gecko may normally exhibit a high variability in testosterone concentrations between individuals, or it is possible that the breeding season of these animals is less well defined than previously thought in captive situations. For example, when in captivity, the Madagascar ground gecko has been found to breed continuously [70,71,72] and male leopard geckos that have had previous sociosexual experiences were found to express higher circulating androgen concentrations than naïve males [68]. Measuring testosterone concentrations over the course of the reproductive cycle will be necessary to determine if leopard geckos have a three-phase reproductive cycle, characterized by active, recrudescent, and quiescent phases, and to better understand comparisons with other species.

Other possibilities for the lack of testosterone stimulation in the leopard geckos receiving PMSG may be that a higher dose or more frequent dosing is required in these animals to elicit an effect. The doses selected for this study, 20 IU and 50 IU, were standardized to animal rather than an IU/kg basis. This was done because hormones tend to flood all available active sites at the level of the tissue, causing a ceiling effect. The dosages selected in this study were thought to be mid-range, with the aim to use the lowest effective dosage of hormone required to elicit an effect. Previous studies evaluating the effects of PMSG in lizards used more frequent dosing (daily or every other day) with shorter durations of administration (2 days to 21 days) [19,21,23,24,30]. However, in a recent study evaluating the effects of hCG administration in veiled chameleons, it was determined that weekly injections of 100, 200, and 300 IU hCG were sufficient to maintain elevated plasma testosterone concentrations over a month-long period [12]. Additionally, in oriental garden lizards, spermatogenesis was not impacted when 5 IU of PMSG, hCG, or a combination of the two hormones were given daily for 10 days, leading the authors to consider the need for higher doses or a longer period of administration [30]. Based on these results, weekly injections of PMSG were selected for use in leopard geckos with the aim of maintaining elevated plasma testosterone concentrations over a longer period of time to be able to more fully assess the impact on spermatogenesis, since it has been determined that spermatogenesis in reptiles may take between 5–8 weeks to complete [60]. Future studies should consider administering higher dosages (e.g., 100 IU/animal) or administering PMSG on a daily or every other day basis over shorter time periods to determine dosing efficacy. Additionally, blood was sampled 7 days following the most recent PMSG injection, which may have been too long of a time period to catch a peak increase in plasma testosterone concentrations following administration of PMSG. Lastly, the failure of PMSG to stimulate additional testosterone production may be due to the possibility that leopard geckos do not follow the one gonadotroph, two-cell theory of reproduction. Based on the results of this study, PMSG has demonstrated its FSH-like effect in this species by stimulating spermatogenesis. However, testosterone concentrations did not increase concurrently as expected. Ultimately, PMSG administration at a time of quiescence and low baseline testosterone concentrations will be needed to more accurately determine the effect of exogenous PMSG on Leydig cell testosterone production to further characterize if leopard geckos are a species that follows the one gonadotroph, two-cell theory.

There were several limitations associated with this study. Our lack of understanding of the leopard gecko reproductive cycle in captivity limited our ability to perform this study during a time of low testosterone production in order to evaluate the effect of PMSG on stimulating testosterone production in this species. Future studies may consider measuring study subject testosterone concentrations, prior to recruitment into the study, to ensure levels are truly associated with a quiescent phase of reproduction. Additionally, more frequent blood collection or sampling within 24 h of PMSG treatment may help identify a peak in circulating levels. Limitations associated with the measurement of testicular volumes via ultrasound were also present in this study. The superimposition of the gastrointestinal tract, urinary bladder, liver, and intra-abdominal fat in the location of the testes made visualization of the testicle borders challenging. Measurement error could have also occurred with the ultrasound unit and caliper placement when measuring the small testes; however, measurements were all performed by the same two authors each time (AM and MM) to limit bias. Another limitation was associated with the suture material used to close the surgical incisions. A quarter (6/24) of the leopard geckos experienced suture reactions to the 4-0 Maxon and required subsequent repair. In all of these cases, the suture material was intact and extruded through the skin or body wall. Based on these findings, the authors do not recommend its use in leopard geckos. Other limitations of this study were associated with the small ejaculate volumes, mechanical losses due to semen evaluation technique, sample contamination, and sperm clumping. Ejaculate volumes of the leopard gecko were similar to those reported in veiled (2.0 µL) and panther chameleons (2.9 µL), but they were lower than those observed in other studies (4.6 µL in the Texas rock lizard, 50 µL in the green iguana) [10,12,15]. The low volume may be attributed to the smaller relative size of the leopard geckos in comparison to the other lizards, in addition to the methodology used to evaluate and collect the samples. Ejaculates were obtained and placed directly on a slide with an overlying cover slip to confirm the presence or absence of spermatozoa and to characterize motility and any contamination present before being washed into a cryovial with 10% buffered formal saline. This step likely led to decreased sample volume for analysis and potentially led to a reduction in spermatozoa concentrations. Additionally, the consistency of white ejaculates was more viscous than samples that were more clear and were found to clump to a higher degree; this also could contribute to falsely lowering concentrations in some samples. Samples were often contaminated with feces and urine, which could have yielded artificial decreases in sperm concentrations. Other studies evaluating lizard spermatozoa used an extender prior to evaluation rather than evaluating raw samples [10,14,15]. A semen extender was not used in this study due to concerns for over-diluting the small semen volumes; however, future studies should consider extenders to increase sample volume and allow for additional sample testing.

## 5. Conclusions

In conclusion, PMSG was found to stimulate spermatogenesis in captive leopard geckos between October–December in the Northern Hemisphere; however, it had no effect on testosterone production during these same months. Sampling time may have affected testosterone concentrations, and more frequent sampling may be necessary. This exogenous gonadotropin was found to increase testicular volume and weights, in addition to sperm concentration and motility. Ultrasound can be used to safely and accurately measure testicular volumes ante-mortem in leopard geckos. Electrostimulation is also an effective tool to collect serial semen samples in the leopard gecko. Future studies are needed to further characterize the annual reproductive cycle of leopard geckos to gain insight into the specific phases of this prenuptial reptile.

## Figures and Tables

**Figure 1 animals-11-02477-f001:**
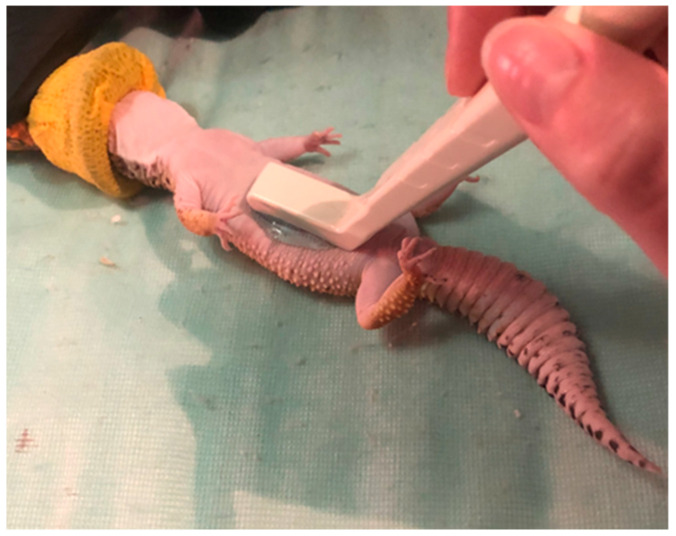
Ultrasounding leopard gecko to measure testicle size.

**Figure 2 animals-11-02477-f002:**
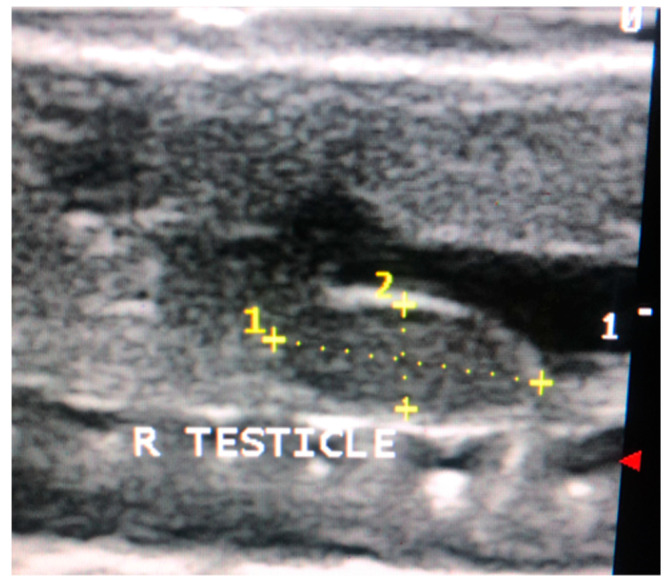
Right testicular measurement on ultrasound.

**Figure 3 animals-11-02477-f003:**
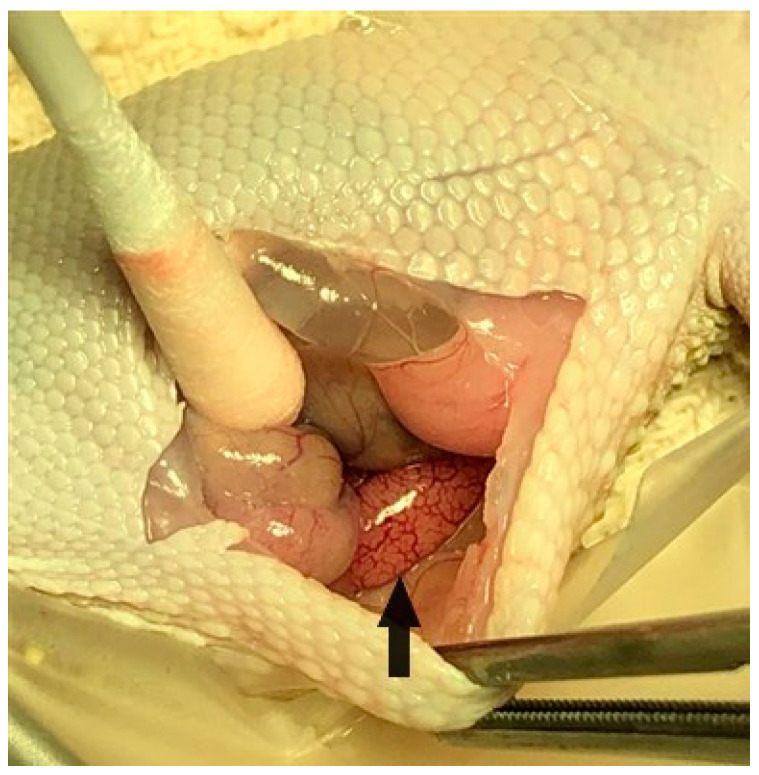
Visualization of right testicle (arrow) in body cavity prior to removal.

**Figure 4 animals-11-02477-f004:**
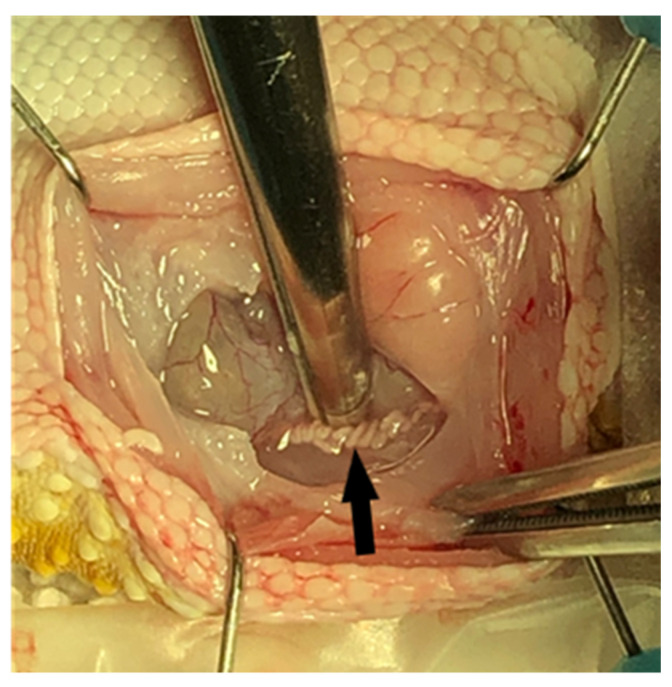
Visualization of right epididymis (arrow) in body cavity prior to removal.

**Figure 5 animals-11-02477-f005:**
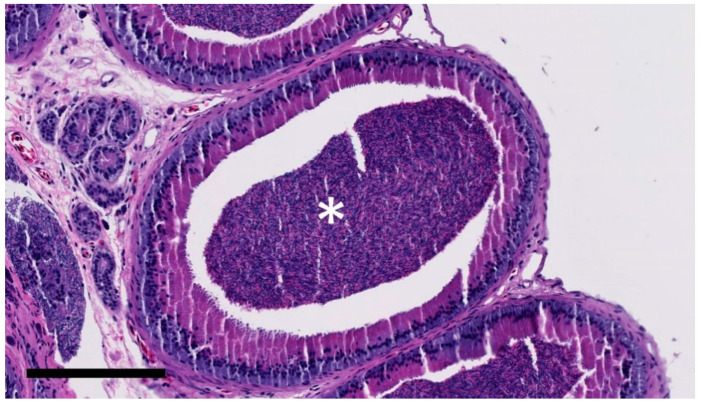
Abundant mature spermatozoa (white asterisk; grade 3/3) in epididymis of gecko receiving PMSG. Bar = 200 μm.

**Figure 6 animals-11-02477-f006:**
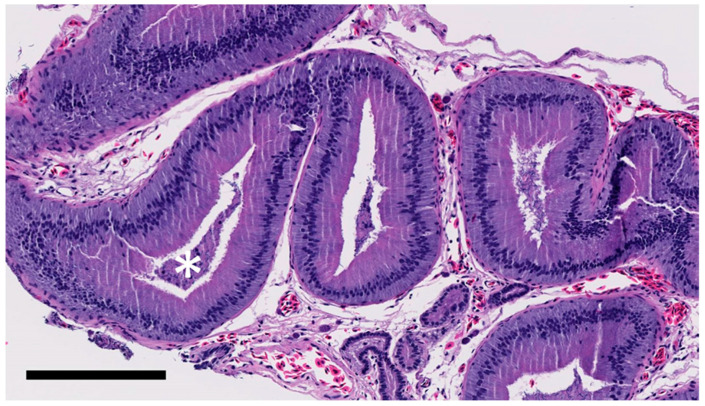
Few mature spermatozoa (white asterisk; grade 1/3) in epididymis of gecko receiving saline. Bar = 200 μm.

**Table 1 animals-11-02477-t001:** Leopard gecko body weights (grams) over time.

Time	*n*	Median	25–75%	Min–Max
Baseline	24	79.6 ^a^	75.6–83.0	46.0–91.0
21 days	24	78.4 ^b^	74.9–82.4	46.8–88.3
42 days	24	77.0 ^c^	74.2–81.7	45.0–87.0
63 days	24	75.7 ^a,b,c^	75.7–81.2	44.5–84.6

^a,b^*p* = 0.0001, ^c^
*p* = 0.012.

**Table 2 animals-11-02477-t002:** Leopard gecko testicular volume (mm^3^) measured by ultrasound over time.

Time	*n*	Median	25–75%	Min–Max
Baseline	24	38.0 ^a^	29.98–64.34	13.05–79.61
21 days	24	52.65 ^c^	32.91–67.52	14.65–122.91
42 days	24	54.01 ^a^	36.89–61.9	15.78–112.05
63 days	24	66.01 ^b,c^	35.36–79.62	20.33–141.52

^a,c^*p* = 0.04, ^b^
*p* < 0.0001.

**Table 3 animals-11-02477-t003:** Leopard gecko post-surgical testicular volume (mm^3^) and weight (grams) by treatment group.

Parameter	Group	*n*	Median	25–75%	Min–Max
Testicular volume					
	Saline	8	66.6 ^a,b^	32.7–117.3	10.40–143.0
	20 IU	8	124.3 ^a^	86.1–168.9	83.2–196.6
	50 IU	8	97.6 ^b^	58.9–161.0	51.5–282.8
Testicular weight					
	Saline	8	58.0 ^c,d^	28.2–96.0	9.0–120.0
	20 IU	8	95.0 ^c^	74.5–118.7	47.0–148.0
	50 IU	8	68.0 ^d^	59.7–112.7	47.0–195.0

^a^ *p* = 0.009, ^b^ *p* = 0.036, ^c^ *p* = 0.018, ^d^ *p* = 0.043.

**Table 4 animals-11-02477-t004:** Leopard gecko spermatozoa motility (%) over time.

Time	*n*	Median	25–75%	Minimum–Maximum
0	24	0 ^a,b^	0–10	0–80
21 days	24	3 ^c,d^	0–10	0–80
42 days	24	25 ^a,c^	5–60	0–90
63 days	24	45 ^b,d^	5–67.5	0–90

^a^*p* = 0.021, ^b^
*p* = 0.01, ^c^
*p* = 0.053, ^d^
*p* = 0.005.

**Table 5 animals-11-02477-t005:** Leopard gecko spermatozoa motility (%) by treatment.

Treatment	*n*	Median	25–75%	Minimum–Maximum
Saline	8	5 ^a,b^	0–21	0–90
20 IU	8	25 ^a^	1–70	0–90
50 IU	8	10 ^b^	0–60	0–90

^a^ *p* = 0.006, ^b^ *p* = 0.049.

**Table 6 animals-11-02477-t006:** Leopard gecko spermatozoa concentrations over time and by treatment group.

Model 1					
Time	Group	*n*	Median	25–75%	Min–Max
Baseline ^a^	Saline	8	2.4 × 10^7^	6.3 × 10^6^–1.4 × 10^8^	1.8 × 10^6^–1.77 × 10^8^
	20 IU	8	1.7 × 10^7^	1.1 × 10^7^–6.8 × 10^7^	1.0 × 10^7^–8.5 × 10^7^
	50 IU	8	3.4 × 10^7^	1.1 × 10^7^–3.4 × 10^8^	9.0 × 10^6^–2.3 × 10^8^
Day 21 ^c^	Saline	8	2.9 × 10^7^	1.7 × 10^7^–1.3 × 10^8^	1.3 × 10^7^–2.1 × 10^8^
	20 IU	8	1.8 × 10^7^	6.2 × 10^6^–7.0 × 10^7^	5.3 × 10^6^–1.4 × 10^8^
	50 IU	8	1.4 × 10^7^	4.7 × 10^6^–7.8 × 10^7^	2.8 × 10^6^–4.9 × 10^8^
Day 42 ^b,d^	Saline	8	9.2 × 10^6^	4.8 × 10^6^–5.6 × 10^7^	2.0 × 10^6^–1.1 × 10^9^
	20 IU	8	3.5 × 10^8^	5.5 × 10^7^–1.0 × 10^9^	5.3 × 10^7^–1.1 × 10^9^
	50 IU	8	2.8 × 10^8^	1.1 × 10^8^–7.1 × 10^8^	2.0 × 10^6^–4.3 × 10^9^
Day 63 ^b,d^	Saline	8	4.4 × 10^7^	1.0 × 10^7^–1.9 × 10^8^	2.8 × 10^6^–1.8 × 10^8^
	20 IU	8	1.2 × 10^8^	2.4 × 10^7^–1.0 × 10^9^	2.1 × 10^7^–2.7 × 10^9^
	50 IU	8	1.1 × 10^8^	6.0 × 10^7^–2.2 × 10^8^	2.0 × 10^7^–8.3 × 10^8^
**Model 2**					
Time	Group	*n*	Median	25–75%	Min–Max
Baseline	Saline	8	2.4 × 10^7^	6.3 × 10^6^–1.4 × 10^8^	1.8 × 10^6^–1.8 × 10^8^
	PMSG	16	2.0 × 10^7^	1.1 × 10^6^–7.2 × 10^8^	9.0 × 10^6^–2.3 × 10^8^
Day 21	Saline	8	2.9 × 10^7^	1.7 × 10^7^–1.4 × 10^8^	1.3 × 10^7^–2.1 × 10^8^
	PMSG	16	1.5 × 10^7^	6.2 × 10^6^–7.0 × 10^7^	2.8 × 10^6^–4.9 × 10^8^
Day 42	Saline	8	9.2 × 10^6^	4.8 × 10^6^–5.6 × 10^7^	2.0 × 10^6^–1.1 × 10^9^
	PMSG	16	3.0 × 10^8^	9.0 × 10^7^–7.5 × 10^8^	2.0 × 10^6^–4.3 × 10^9^
Day 63	Saline	8	4.4 × 10^7^	1.0 × 10^7^–1.9 × 10^8^	2.8 × 10^6^–2.4 × 10^8^
	PMSG	16	1.1 × 10^8^	3.6 × 10^7^–4.9 × 10^8^	2.0 × 10^7^–2.7 × 10^9^

^a,b^ *p* < 0.009, ^c,d^ *p* < 0.004.

**Table 7 animals-11-02477-t007:** Differences in normal spermatozoa morphology (% total morphology) by treatment group.

Treatment Group	*n*	Mean	SD	Min–Max
Saline	8	17.4	8.0	4–36
20 IU PMSG	8	28.5	15.0	4–60
50 IU PMSG	8	22.0	10.5	4–44

**Table 8 animals-11-02477-t008:** Abnormal spermatozoa morphology (% total morphology) that differed over time.

Abnormality	Time	*n*	Median	25–75%	Min–Max
Folded tail	Baseline	24	26.0	21.5–40	14–60
	21 days	24	30.0	20–36	14–60
	42 days	24	37.0	27–41	25–46
	63 days	24	45.5	40–49.5	17–72
Kinked midbody	Baseline	24	19.0	10–26.7	2–34
	21 days	24	18.0	14–25	2–36
	42 days	24	12.0	3–15	0–24
	63 days	24	7.5	5.7–13.7	0–24

**Table 9 animals-11-02477-t009:** Abnormal spermatozoa morphology (% total morphology) that was not different by time or treatment group.

Abnormality	*n*	Median	25–75%	Min–Max
Distal droplet	24	0	0–0	0–12
Head defect	24	0	0–1.7	0–9
Detached head	24	0	0–2	0–12
Proximal droplet	24	12	7–16	0–33
Coiled tail	24	6	2–13.7	0–56

**Table 10 animals-11-02477-t010:** Leopard gecko testosterone concentration reference intervals for October, November, and December in the Northern hemisphere. Testosterone concentrations were not found to meet the assumption of normality and are reported by the median, 10–90%, minimum–maximum values, 95% reference intervals, using both the robust and nonparametric methods, and the 90% confidence intervals for the upper and lower limits of the reference interval nonparametric and minimum–maximum values. The nonparametric data are preferred for these reference intervals.

Parameter	Median	Min–Max	10–90%	95th Percentile Reference Interval (Robust Method)	90% CI for Lower Limit	90% CI for Upper Limit	95th Percentile Reference Interval (Non-Parametric)	90% CI for Lower Limit	90% CI for Upper Limit
Testosterone October (ng/mL) (*n* = 22)	82.1	29.5–216.12	32.0–287.2	−47.2–206.3	−76.4–−10.2	163.3–259.1	29.5–216.1	N/A	N/A
Testosterone November (ng/mL) (*n* = 24)	77.2	17.5–465.7	22.6–327.7	−195.2–388.3	−276.3–−82.7	264.4–474.5	17.5–465.7	N/A	N/A
Testosterone December (ng/mL) (*n* = 24)	79.0	11.1–399.8	25.5–350.2	−183.4–340.6	−247.1–−82.2	244.2–441.8	11.1–399.8	N/A	N/A

## Data Availability

The data presented in this study are available on request from the corresponding author.

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
