# Peer review of "Determining the Effects of Serial Injections of Pregnant Mare Serum Gonadotropin on Plasma Testosterone Concentrations, Testicular Dynamics, and Semen Production in Leopard Geckos (Eublepharis macularius)"

_animals, 2021, doi:10.3390/ani11092477_

Round 1
Reviewer 1 Report
The work developed presents originality and scientific significance, is well presented and discussed, however, although it has shown the great susceptibility of reptiles to extinction, the authors did not make clear the importance of studying the reproduction of Eublepharis macularius (is it an endangered species?; is it easy to obtain and maintain in captivity? how the information obtained in this study can contribute to the development of breeding programs for endangered species of lizards???).
In the conclusions, it would be important to clarify that the non-effect of the PMSG on testosterone production may have resulted from flaws in the methodology for collecting blood samples, which needs tde adjustments so that peaks in circulating testosterone levels can be identified.
Although some references are very old (26%), they were used to support revisional data and do not compromise the basis of the discussions.
Author Response
Reviewer 1
The work developed presents originality and scientific significance, is well presented and discussed, however, although it has shown the great susceptibility of reptiles to extinction, the authors did not make clear the importance of studying the reproduction of Eublepharis macularius (is it an endangered species?; is it easy to obtain and maintain in captivity? how the information obtained in this study can contribute to the development of breeding programs for endangered species of lizards???).
Thank you for this excellent suggestion. A sentence has now been added (lines 127-130) that reads as follows: “The leopard gecko was chosen as a model gecko species due to their abundance, ease of maintaining in captivity and previously established reproductive seasonality [40]. Refining these techniques in a common species will be important before applying them to threatened or endangered species. ”
In the conclusions, it would be important to clarify that the non-effect of the PMSG on testosterone production may have resulted from flaws in the methodology for collecting blood samples, which needs tde adjustments so that peaks in circulating testosterone levels can be identified.
Thank you for this suggestion. We have specifically added the following statement to the conclusions: lines 927-928: “Sampling time may have affected testosterone concentrations and more frequent sampling may be necessary.”. Additionally, the following was already addressed in the discussion, which we believe reinforces your concern: lines 889-892: “ Future studies may consider measuring study subject testosterone concentrations prior to recruitment into the study to ensure levels are truly associated with a quiescent phase of reproduction. Additionally, more frequent blood collection or sampling within 24 hours of PMSG treatment may help identify a peak in circulating levels”, and lines 844-847: “ Additionally, blood was sampled 7 days following the most recent PMSG injection, which may have been too long of a time period to catch a peak increase in plasma testosterone concentrations following administration of PMSG.”
Although some references are very old (26%), they were used to support revisional data and do not compromise the basis of the discussions.
Thank you for noting this finding. This should reinforce the importance of this paper. To conserve the 300 million years of evolution these species have developed, we must identify ways to save them. Unfortunately, this field has had little focus and the older references are some of the few that are available. We hope to be part of this change with the work going on in our lab.
Reviewer 2 Report
This manuscript used pregnant mare serum gonadotropin to stimulate the production of sperm and increase their testosterone concentrations in leopard geckos. The results of this study confirmed the majority of the authors’ original hypotheses, except two: that animals administered PMSG would have histological changes associated with increased testosterone production and that there would be higher circulating plasma testosterone concentrations in PMSG treated animals compared to controls. The authors may should discuss whether the main androgen hormone in this animla is exactly testosterone or possible others. And also, several concerns are following.
- Body weights are consistently reduced means the experimental animals in stress, how to resolve this problem, please try to explain more.
- From testicular weights, 20 IU PMSG looks better than 50 IU, however, other items are different, please give some reasons to explian.
- Please put bars into figures 5 and 6.
- Please put some scale rulers into figures 2-4.
Author Response
Reviewer 2
This manuscript used pregnant mare serum gonadotropin to stimulate the production of sperm and increase their testosterone concentrations in leopard geckos. The results of this study confirmed the majority of the authors’ original hypotheses, except two: that animals administered PMSG would have histological changes associated with increased testosterone production and that there would be higher circulating plasma testosterone concentrations in PMSG treated animals compared to controls. The authors may should discuss whether the main androgen hormone in this animla is exactly testosterone or possible others. And also, several concerns are following.
- Body weights are consistently reduced means the experimental animals in stress, how to resolve this problem, please try to explain more.
Thank you for this comment. The authors attempted to exhaustively address the decrease in body weight in the second paragraph of the discussion and listed physiologic stress (lines 613-615), frequency of handling (lines 615-617), increased locomotion related to reproductive activity (lines 618-622), reduced food consumption during periods of breeding (lines 618-622), prenuptial reproductive cycle (lines 623-624), and differences in tail lengths (lines 624-628) as potential factors that could have played a role in this decrease. We specially addressed mitigation of stress in lines 617-618:” It is possible that a shorter study with more frequent dosing could reduce overall handling and should be considered in the future.” To further address loss from energy availability we have now also added the following: lines 624-628: “Increasing the availability of food during periods of increased reproductive activity may help offset weight loss; however, if these animals have reduced food consumption for physiologic reasons, we may just need to expect weight loss during this period of the reproductive cycle.”
- From testicular weights, 20 IU PMSG looks better than 50 IU, however, other items are different, please give some reasons to explian.
Thank you for this comment. Because this was not a significant finding (lines 406-407, 417-428, Table 2) you should not look at them as different (or better). As mentioned in the discussion, a dose dependent response for the effect of PMSG on varying parameters in the leopard geckos was not found as was noted in the following sentence (lines 650-651) “Thus, the effects of PMSG in lizards may not be dose dependent.” This finding was consistent with other references cited in the paper. Additionally, we addressed why testicular volumes were different between PMSG and saline groups in lines 632-642: “PMSG increased testicular sizes in treatment animals compared to those administered saline, as measured by elevated GSI, testicular weights, ultrasound and post-operative testicular volumes, and correlating with a higher degree of sperm production in PMSG animals. Spermatozoa are produced in the testicles and testis size in reptiles is maximal at the time of spermiogenesis, suggesting that large testes are indicative of a high spermatozoa production at the individual level [51]. In the common agama, a four-fold increase in GSI (mean GSI 0.88) was observed in animals receiving PMSG compared to control animals after 21 days [21]. While the difference in GSI was not as dramatic in the leopard geckos, a nearly two-fold increase was observed in animals administered PMSG compared to controls, suggesting that PMSG administration had a significant impact on spermatogenesis.”
- Please put bars into figures 5 and 6.
Thank you for this suggestion. We have added the measurement bars.
- Please put some scale rulers into figures 2-4.
Because these are photographs (and we don’t have measurements built in) we cannot add rulers. Sorry.
Reviewer 3 Report
The manuscript has very well written sections, I would even say that the introduction, some parts of the material and methods section, as well as the discussion are exhaustively described and with very rigorous details. It is a very elaborate manuscript, with a lot of important information that can be used by other researchers working with these species, so I think it deserves to be published. However, before publishing it they should mainly improve the results section.
Before commenting on specific aspects of the manuscript, I would like to point out that in recent years the acronym PMSG (Pregnant mare serim gonadotrphin) has tended to be replaced by the acronym eCG (equine chorionic gondotrophin). Thus the nomenclature is similar to hCG.
In the introduction:
You must add references on lines 98-100.
You must explain why you choose serial injections and not a single injection.
Doses of 20 and 50 IU per animal, in my opinion, are very high. For example, a single dose of 25 IU for a female rabbit is effectively used and causes sufficient ovarian stimulation. Keep in mind that a rabbit weighs 4-5 kg !!
Be careful with the superscripts to indicate millions of sperm (for example: lines 154-155) or to indicate degrees of temperature (line 160). Review the rest of the manuscript.
In material and methods:
This section is very well described. All procedures are very well explained (I would even say too exhaustively), for example, when they indicate the name of the forceps and all the surgical material used.
However, in my opinion, they should indicate the age of the animals. This is a very important parameter and I also think it would be interesting to know if these animals have reproduced previously.
I don't understand why authors used these animals in the non-breeding season.
Regarding the determination of sperm motility or the histological study, authors must indicate whether it was always determined by the same technician.
I also find it strange that to analyse testosterone, blood samples do not undergo an extraction procedure. In other species, it is usually done.
In results:
In the figures, it is necessary to indicate with arrows or other symbols what the reader should look at. For example, in figure 4 (above) there will be readers who cannot distinguish between fat and testicle, or in figure 4 (below) the reader does not know what he has to see in that photo.
The same, for the histological sections in which I advise you to indicate where the seminiferous tubes are, the sperm, the epithelium (thickness), the interstitial tissue, etc. It would also be convenient if they provided images of the most important morphological abnormalities found in sperm. I consider these images to be more important than the photo of the lizard anesthetized with the ultrasound probe.
Much of the determined parameters could be included in a single table. For example, the animal live body weight, the volume of the testicles (measured by ultrasound), as well as the weight and volume of the extracted testicle could all be included in a single table. Normally, the rows of the table indicate the means of the parameters studied (weight, volume ...) and the columns indicate the effect studied (time and treatment), as well as the interaction of both. Only if the interaction is significant, a figure is made showing how the parameter in question varies depending on the treatment and time.
I believe that it is not necessary to give the maximum and minimum values, but rather that it is better that they provide the standard deviation of the mean of each group. The letters written above the mean values should indicate statistically significant differences. Different letters are usually written to indicate significant differences between groups, and the authors of this manuscript do the opposite. Also, the authors do not put the units in any result of weight, volume ... and this is very important so that the tables can be understood.
Regarding testosterone concentrations, high variability between animals is observed that prevents significant differences between groups. This variability could indicate that the number of animals per group should be increased. This variability could be also due to the age or sexual maturity of the animals.
How do you explain that spermatogenesis increases in the groups treated with PMSG without an increase in testosterone, if we know that this hormone, together with the binding protein, is responsible for stimulating this process in the seminiferous epithelium?
Finally, the discussion of the results is very thorough and correct. It is even very interesting that the authors indicate the drawbacks of their study and try to provide explanations for the possible errors made.
Author Response
Reviewer 3
The manuscript has very well written sections, I would even say that the introduction, some parts of the material and methods section, as well as the discussion are exhaustively described and with very rigorous details. It is a very elaborate manuscript, with a lot of important information that can be used by other researchers working with these species, so I think it deserves to be published. However, before publishing it they should mainly improve the results section.
Before commenting on specific aspects of the manuscript, I would like to point out that in recent years the acronym PMSG (Pregnant mare serim gonadotrphin) has tended to be replaced by the acronym eCG (equine chorionic gondotrophin). Thus the nomenclature is similar to hCG.
Thank you for your kind words. We kept the PMSG for consistency with the reptile literature. We leave this up to the reviewer and editor. If your preference is to change PMSG to eCG throughout, we will do it. However, since the field is still referring to PMSG, and we purchased the drug as PMSG (lines 177-179: PMSG (Pregnant mare serum gonadotropin, sterile filtered white lypolized powder, 1,000IU, ProSpec-Tany TechnoloGene LTD, Rehovot, Israel)), our preference it to leave it as is.
In the introduction:
You must add references on lines 98-100.
Thank you. References added [21,25,26, 28-29, 31,34-38] .
You must explain why you choose serial injections and not a single injection.
Thank you for this comment. The following sentence can be found in the discussion that should address the authors’ reasoning for performing serial rather than a single injection of PMSG: lines 872-876: “…weekly injections of PMSG were selected for use in leopard geckos with the aim of maintaining elevated plasma testosterone concentrations over a longer period of time to be able to more fully assess the impact on spermatogenesis, since it has been determined that spermatogenesis in reptiles may take between 5-8 weeks to complete [60].”
Doses of 20 and 50 IU per animal, in my opinion, are very high. For example, a single dose of 25 IU for a female rabbit is effectively used and causes sufficient ovarian stimulation. Keep in mind that a rabbit weighs 4-5 kg !!
Thank you for this comment. Of course, differences have been found between species across classes, so we should not be surprised to see the variability. To date, there is no standardized dosing for this hormone in reptiles; however, doses used in reptiles are higher than those for mammals. We addressed this in lines 648-653: “Effective dosages of PMSG in lizards have ranged from 1 IU in Leiolopisma laterale [24] to 100 IU in the Agama agama [21]. Thus, the effects of PMSG in lizards may not be dose dependent. While no current attempt has been made to standardize dosing, standardization will be necessary in order to develop functional reproductive programs in the future.” Additionally, we further commented on this in lines: 860-872- “The doses selected for this study, 20IU and 50IU, were standardized to animal rather than an IU/kg basis. This was done because hormones tend to flood all available active sites at the level of the tissue, causing a ceiling effect. The dosages selected in this study were thought to be mid-range, with the aim to use the lowest effective dosage of hormone required to elicit an effect. Previous studies evaluating the effects of PMSG in lizards used more frequent dosing (daily or every other day) with shorter durations of administration (2 days to 21 days) [19,21,23,24,30]. However, in a recent study evaluating the effects of hCG administration in veiled chameleons, it was determined that weekly injections of hCG were sufficient to maintain elevated plasma testosterone concentrations over a month-long period [12]. “ In the last study, the authors used 100, 200, and 300 IU IU. We have added that to line, it now reads: Lines 866-869: However, in a recent study evaluating the effects of hCG administration in veiled chameleons, it was determined that weekly injections of 100, 200, ad 300 IU hCG were sufficient to maintain elevated plasma testosterone concentrations over a month-long period [12].
Finally, we referenced our species doses based on three specific articles: Lines 178-181-The chosen dosages of PMSG (Pregnant mare serum gonadotropin, sterile filtered white lypolized powder, 1,000IU, ProSpec-Tany TechnoloGene LTD, Rehovot, Israel) were based on previous work performed in reptiles [19,23,39].
Be careful with the superscripts to indicate millions of sperm (for example: lines 154-155) or to indicate degrees of temperature (line 160). Review the rest of the manuscript.
Thank you. We’re not sure what happened. When we uploaded the MS, all superscripts were present and Latin names italicized. Opening the file from the Animals systems seems to have altered this. We have corrected.
In material and methods:
This section is very well described. All procedures are very well explained (I would even say too exhaustively), for example, when they indicate the name of the forceps and all the surgical material used.
However, in my opinion, they should indicate the age of the animals. This is a very important parameter and I also think it would be interesting to know if these animals have reproduced previously.
Thank you. The authors’ felt it was important for completeness to include as much detail is possible so that the study could be effectively repeated by others if desired. In terms of the animals’ history, a sentence has now been added to reflect this and reads as follows (Lines 153-156): “The ages and previous sexual histories of the male geckos used in this study are unknown, however, no male in this study had previous contact with females for at least three years while housed at LSU.”
I don't understand why authors used these animals in the non-breeding season.
Thank you. The purpose of this study was to establish if PMSG was able to increase plasma testosterone concentrations and stimulate spermatogenesis in a model species of gecko. In order to evaluate this, the authors aimed to conduct the study during a time of reproductive quiescence to see if this exogenous hormone could induce spermatogenesis and increases in circulating testosterone. The thought being to determine if exogenous PMSG administration can be used to cycle geckos to breed during times of quiescence to gain a better understanding of their reproductive physiology that can then be applied in the development of assisted reproductive technologies in other threatened and endangered species in the future. However, the geckos in this study already had elevated testosterone concentrations and many were able to have semen collected regardless of PMSG administration which is what led to the proposal that leopard geckos in captivity in the Northern hemisphere may follow a three-phase reproductive cycle with a recrudescent phase. Future studies are needed to further characterize the annual reproductive cycle of leopard geckos to gain insight into the specific phases of their reproductive cycle to then have a better understanding of what PMSG can do during sexual quiescence. We addressed this in the discussion (lines 844-860).
Regarding the determination of sperm motility or the histological study, authors must indicate whether it was always determined by the same technician.
Thank you. Sentences have now been added and read as follows: Lines 244-245: “All motilities were measured by a single reviewer (MM)”, Lines 254-255: “All spermatozoa counts and morphology were performed by a single reviewer (AM)”, and Line 325: “All samples were reviewed by the same author (JL).”
I also find it strange that to analyse testosterone, blood samples do not undergo an extraction procedure. In other species, it is usually done.
Thank you. Rather than meticulously listing the steps taken for the testosterone assay, the methods for the current paper state that: “This assay has been previously validated by the authors in leopard geckos, and the same methods were followed in the present study [32].” As part of this separate study, it is noted in this paper that “Plasma testosterone was extracted using a liquid extraction method based on the manufacturer’s protocol.”
In results:
In the figures, it is necessary to indicate with arrows or other symbols what the reader should look at. For example, in figure 4 (above) there will be readers who cannot distinguish between fat and testicle, or in figure 4 (below) the reader does not know what he has to see in that photo.
Thank you. The following have been added:
Line 299: Figure 3. Visualization of right testicle (arrow) in body cavity prior to removal.
Line 301: Figure 4. Visualization of right epididymis (arrow) in body cavity prior to removal.
The same, for the histological sections in which I advise you to indicate where the seminiferous tubes are, the sperm, the epithelium (thickness), the interstitial tissue, etc. It would also be convenient if they provided images of the most important morphological abnormalities found in sperm. I consider these images to be more important than the photo of the lizard anesthetized with the ultrasound probe.
Thank you. The focus of these two images was to reinforce the spermatozoa counts (see figure legends). We selected these images because they show the variability (3/3 v 1/3) between groups. We have added the measurement bars and white asterisks to characterize spermatozoa in images. We do not feel it is appropriate to identify all other sections, as they are not relevant to the focus of this image. We did not include images of all differences because of length of paper (already quite long). We hope this suffices. At some point readers need to understand what images represent. In looking through other Animal articles with histopath, only tissues identified in legends are keyed in images, so that’s what we did. Unfortunately, we do not have good images of the sperm morphology. We do think the gecko U/S image is important for the reader to reinforce even though these lizards are small, this diagnostic test can be powerful and useful. Many don’t realize they can use such a tool on a small animal.
Much of the determined parameters could be included in a single table. For example, the animal live body weight, the volume of the testicles (measured by ultrasound), as well as the weight and volume of the extracted testicle could all be included in a single table. Normally, the rows of the table indicate the means of the parameters studied (weight, volume ...) and the columns indicate the effect studied (time and treatment), as well as the interaction of both. Only if the interaction is significant, a figure is made showing how the parameter in question varies depending on the treatment and time. I believe that it is not necessary to give the maximum and minimum values, but rather that it is better that they provide the standard deviation of the mean of each group. The letters written above the mean values should indicate statistically significant differences. Different letters are usually written to indicate significant differences between groups, and the authors of this manuscript do the opposite. Also, the authors do not put the units in any result of weight, volume ... and this is very important so that the tables can be understood.
Thank you, but we disagree. The senior author is an epidemiologist and has concerns about how data are presented in most papers. We detailed the process of analyzing the data, including distributions, to determine best methods for reporting central tendency and variability. We must use these tools to ensure we are providing “best practices”. Maximum-minimum values are essential to recognizing the range of values and provide the reader an objective idea of the dispersion of the data. The tables are also done individually to match with respective paragraphs to simplfy review for the readers. We don’t understand the concern for the superscript call outs to signify significance. This is a common practice. The senior author has used this method for some 200+ papers, and routinely sees them in the Journals he serves(d) as editor in chief. If the Editor prefers a different method, we will change it.
Excellent point about Tables not having methods of measurements in Tables. All have been added where they were missing (Tables 1- 5). Thank you!
Regarding testosterone concentrations, high variability between animals is observed that prevents significant differences between groups. This variability could indicate that the number of animals per group should be increased. This variability could be also due to the age or sexual maturity of the animals. How do you explain that spermatogenesis increases in the groups treated with PMSG without an increase in testosterone, if we know that this hormone, together with the binding protein, is responsible for stimulating this process in the seminiferous epithelium?
Thank you. We agree that there is high variability in testosterone, but don’t believe a larger sample size will help this. We addressed this by acknowledging these animals were likely in a recrudescent phase, similar to some other gecko species. A better comparison would be do look outside the months we evaluated, but that is for another study (see lines 860-863 and : “Measuring testosterone concentrations over the course of the reproductive cycle will be necessary to determine if leopard geckos have a three-phase reproductive cycle, characterized by active, recrudescent, and quiescent phases, and to better understand comparisons with other species.” And “Ultimately, PMSG administration at a time of quiescence and low baseline testosterone concentrations will be needed to more accurately determine the effect of exogenous PMSG on Leydig cell testosterone production to further characterize if leopard geckos are a species that follows the one gonadotroph, two-cell theory.” Additionally, while we performed the study outside the captive breeding season, nobody had ever assessed testosterone in this species during this phase and we believe we can now say confidently these animals are producing testosterone during this phase (pre-nuptial), which primes them for spermatogenesis; thus, this is why the PMSG groups produced more sperm then the controls (sperm counts and histo results). This is addressed in Lines:783-837.
Finally, the discussion of the results is very thorough and correct. It is even very interesting that the authors indicate the drawbacks of their study and try to provide explanations for the possible errors made.
Thank you!
Round 2
Reviewer 3 Report
I appreciate the very clear and concrete answers to my questions and accept most of your comments on my suggestions, including the reasons for not changing the name of the hormone used in the treatments (PMSG vs eCG).
However, I have to disagree with the answer on how to display the results in the tables. I can admit that you want to show the maximum and minimum values to recognize the range of values and to provide the reader with an objective idea of the spread of the data. Perfect. However, I do not agree with the way you use superscripts to indicate the differences between the means. I also work as an editor of scientific journals, I review more than 10 manuscripts every week and I do not accept as correct this way of indicating differences between the means of the values studied. For example, in the first table, if at 63 days the body weight of the lizards was significantly (p from 0.012 to 0.0001) lower than at the rest of the moments, the mean at 63 days should indicate a superscript different from the rest. It would be like this:
79.6 a
78.4 a
77.0 a
75.7 b;
Different letters indicate significant differences between means (a, b: p <0.05) (it is written p <0.05 because the highest P is p = 0.012).
Another important thing for the results in the tables to be well understood is to indicate the number of animals (data) used to calculate the means.
The editor-in-chief of Animals could accept this way of describing the results because fortunately each table is very well explained in its corresponding paragraph. But please, do not try to convince me that this is the most common way to present the results because it is absolutely not true.
Nor do I accept that they prepare a table to show the results of each parameter they have measured (body weight, testicular weight and volume), they should make the effort to make a common table with the results shown in tables 1, 2 and 3.
Author Response
I appreciate the very clear and concrete answers to my questions and accept most of your comments on my suggestions, including the reasons for not changing the name of the hormone used in the treatments (PMSG vs eCG).
However, I have to disagree with the answer on how to display the results in the tables. I can admit that you want to show the maximum and minimum values to recognize the range of values and to provide the reader with an objective idea of the spread of the data. Perfect. However, I do not agree with the way you use superscripts to indicate the differences between the means. I also work as an editor of scientific journals, I review more than 10 manuscripts every week and I do not accept as correct this way of indicating differences between the means of the values studied. For example, in the first table, if at 63 days the body weight of the lizards was significantly (p from 0.012 to 0.0001) lower than at the rest of the moments, the mean at 63 days should indicate a superscript different from the rest. It would be like this:
79.6 a
78.4 a
77.0 a
75.7 b;
Different letters indicate significant differences between means (a, b: p <0.05) (it is written p <0.05 because the highest P is p = 0.012).
Another important thing for the results in the tables to be well understood is to indicate the number of animals (data) used to calculate the means.
The editor-in-chief of Animals could accept this way of describing the results because fortunately each table is very well explained in its corresponding paragraph. But please, do not try to convince me that this is the most common way to present the results because it is absolutely not true.
Nor do I accept that they prepare a table to show the results of each parameter they have measured (body weight, testicular weight and volume), they should make the effort to make a common table with the results shown in tables 1, 2 and 3.
We want to again thank reviewer 3 for taking the time to read our edited manuscript and we apologize if our response suggested there is only one way to set up tables. This is a long standing concern for the corresponding author. As scientists, we should strive for clear-cut, objective methods; however, for tables, references, etc., every journal has their own twist on these subjects. This should be standardized, but we are as subject to politics in our field as those that run our respective countries. I hope we can move forward and set an international standard. While we also understand journals certainly present data in Tables as you note, not sharing the actual p values short changes the reader and interpretation of the result. Including a p =0.012 to those that are P<0.001 as a p<0.05 hides the magnitude of the difference between the comparisons. A p<0.05 is something that has become a dogma that has led to many misinterpetations of data. We must re-evaluate this, and it is one reason we believe so strongly in keeping the specific p values. By demonstrating the superscript letter between two medians, the reader can see that the values are different and the magnitude of the difference. In Table 1, the differences from baseline and 21 days are much more significant from 63 days and should be identified as such. Major differences in p values were found and reported throughout the Tables, and we believe not showing these differences reduces the understanding of how big some of these differences are. Additionally, we request Tables 1-3 remain independent because they demonstrate very different concepts and because Table 3 is based on treatment groups rather than time (making them more challening to combine because of different independent variables being assessed). We have addedd a column (n) for sample size based on your excellent suggestion to Tables 1-9 (Table 10 already had the sample sizes defined). As the reviewer notes, if the editor wants the changes to the tables, even though we might disagree, we understand the current subjectivity in presenting data; however, we do hope to be part of a continuing change and that the journal Animals can be a part of this change.
